# A Flexible Framework for Discovering Novel Categories with Contrastive Learning

## Abstract

This paper studies the problem of novel category discovery on single- and multi-modal data with labels from different but relevant categories. We present a generic, end-to-end framework to jointly learn a reliable representation and assign clusters to unlabelled data. To avoid over-fitting the learnt embedding to labelled data, we take inspiration from self-supervised representation learning by noise-contrastive estimation and extend it to jointly handle labelled and unlabelled data. In particular, we proposed using category discrimination on labelled data and cross-modal discrimination on multi-modal data to augment instance discrimination used in conventional contrastive learning approaches. We further employ Winner-Take-All (WTA) hashing algorithm on the shared representation space to generate pairwise pseudo labels for unlabelled data to better predict cluster assignments. We thoroughly evaluate our framework on large-scale multi-modal video benchmarks Kinetics-400 and VGG-Sound, and image benchmarks CIFAR10, CIFAR100 and ImageNet, obtaining state-of-the-art results.

## 1 Introduction

With the tremendous advances in deep learning, recent machine learning models have shown superior performance on many tasks, such as image recognition (Deng et al., 2009; Kuznetsova et al., 2020), object detection (Zhou et al., 2019; Tan et al., 2020), image segmentation (Cheng et al., 2019), etc. While the state-of-the-art models might even outperform human in these tasks, the success of these models heavily relies on the huge amount of data with human annotations under the closed-world assumption. Applying deep learning in real (open) world brings many new challenges: it is cost-inhibitive to identify and annotate all categories, and new categories could keep emerging. Conventional methods struggle on handling unlabelled data from new categories (Fontanel et al., 2020). On the flip side, real world provides rich unlabeled data, which are often multi-modal (e.g., video and audio), allowing more possibilities for machine learning models to learn in a similar way as human. Indeed, human learn from multi-modal data everyday with text, videos, audios, etc.

In this paper, we focus on automatically learning to discover new categories in the open world setting. Similar to recent work (Han et al., 2019; 2020) which transfer knowledge from labelled images of a few classes to other unlabelled image collections, we formulate the problem as partitioning unlabelled data from unknown categories into proper semantic groups, while some labelled data from other categories are available. This is a more realistic setting than pure unsupervised clustering which may produce equally valid data partitions following different unconstrained criteria (e.g., images can be clustered by texture, color, illumination, etc) and closed-world recognition which can not handle unlabelled data from new categories without any labels. Meanwhile, our setting is more similar to the human cognition process where humans can easily learn the concept of a new object by transferring knowledge from known objects.

Specifically, we introduce a flexible end-to-end framework to discover categories in unlabelled data, with the goal of utilizing both labelled and unlabelled data to build unbiased feature representation, while transferring more knowledge from labelled to unlabelled data. In particular, we extend the conventional contrastive learning (Chen et al., 2020b; He et al., 2020) to consider both instance discrimination and category discrimination to learn a reliable feature representation on labelled and unlabelled data. We also demonstrate that the cross-modal discrimination would further benefit representation learning on data with multi-modalities. To leverage more of unlabelled data, we

introduce the Winner-Take-All (WTA) hashing (Yagnik et al., 2011) on the shared representation spaces to generate pair-wise pseudo labels on-the-fly, which is the key for robust knowledge transfer from the labelled data to unlabelled data. With the weak pseudo labels, the model can be trained with a simple binary cross-entropy loss on the unlabelled data together with the standard cross-entropy loss on the labelled data. This way our model can simultaneously learn feature representation and performs the cluster assignment using an unified loss function.

The main contributions of the paper can be summarized as follows: (1) we propose a generic, end-to-end framework for novel category discovery that can be trained jointly on labelled and unlabelled data; (2) to the best of our knowledge, we are the first to extend contrastive learning in novel category discovery task by category discrimination on labelled data and cross-modal discrimination on multi-modal data ; (3) we propose a strategy to employ WTA hashing on the shared representation space of both labelled and unlabelled data to generate additional (pseudo) supervision on unlabeled data; and (4) we thoroughly evaluate our end-to-end framework on challenging large scale multi-modal video benchmarks and single-modal image benchmarks, outperforming existing methods by a significant margin.

## 2 RELATED WORK

Our method is related to self-supervised learning, semi-supervised learning, and clustering, while different from each of them. We review the most relevant works below.

Self-supervised learning aims at learning reliable feature representations using the data itself to provide supervision signals during training. Many pretext tasks (e.g., relative position (Doersch et al., 2015), colorization (Zhang et al., 2017), rotation prediction (Gidaris et al., 2018)) have been proposed for self-supervised learning, showing promising results. Recently, the constrastive learning based methods, such as (He et al., 2020) and (Chen et al., 2020b), have attracted lots of attention by its simplicity and effectiveness. The key idea of contrastive learning is *instance discrimination*, i.e., pulling similar pairs close and pushing dissimilar pairs away in the feature space. (Khosla et al., 2020) studied the supervised contrastive learning on labelled data as an alternative of cross-entropy. With the labels, more positive pairs can be generated from the intra-class instances, enabling *category discrimination*. Noise-Contrastive Estimation (NCE) (Gutmann & Hyvärinen, 2010; van den Oord et al., 2018) is an effective contrastive loss widely used in these methods. When handling multi-modal data like videos, different self-supervised learning methods have been proposed to exploit data of different modalities, such as (Patrick et al., 2020; Alayrac et al., 2020; Asano et al., 2020; Alwassel et al., 2019; Morgado et al., 2020). Among them, (Morgado et al., 2020) suggests that cross-modal discrimination can be adopted to improve the representation learning for downstream tasks like image recognition and object detection, which implies that good representations are shared between multiple views of the world. Tian et al. (2019) shows that cross-view prediction outperforms conventional alternatives in contrastive learning on images, depth, video and flow, and more views can lead to better representation. In this paper, we consider the visual and audio modalities for cross-modal learning in videos, and present a new way to incorporate contrastive learning for both labelled and unlabelled data to bootstrap representation learning for novel category discovery.

Semi-supervised learning (Chapelle et al., 2006) considers the setting with labelled and unlabelled data. Specifically, the unlabelled data are assumed to come from the same classes as the labelled data. The objective is to learn a robust model making use of both labelled and unlabelled data to avoid over-fitting to the labelled data. While this problem is well studied in the literature (e.g., (Oliver et al., 2018; Tarvainen & Valpola, 2017; Rebuffi et al., 2020a)), existing methods can not handle unlabelled data from new classes. In contrast, our method is designed to discover new categories in the unlabelled data automatically.

Clustering, which aims at automatically partitioning the unlabelled data into different groups, has long been studied in the machine learning community. There are many classic methods (e.g., $k$-means (MacQueen, 1967), mean-shift (Comaniciu & Meer, 1979)) and deep learning based methods (e.g., (Xie et al., 2016; Dizaji et al., 2017; Rebuffi et al., 2020b)) showing promising results. However, the definition of a cluster can be intrinsically ambiguous, because different criteria can be used to cluster the data. For example, objects can be clustered by color, shape or texture, and the clustering results will be different by taking different criteria, while these criteria cannot be predefined in the

clustering methods. In this paper, we aim to learn these criteria implicitly from the labelled data and transfer them to the unlabelled data on-the-fly.

Until recently, the problem of discovering new categories in unlabelled data by exploiting the labelled data starts to be considered. (Han et al., 2019) introduces a method to first pretrain the model on labelled data followed by fine-tuning with an unsupervised clustering loss. (Han et al., 2020) presents a three-stage method for new image category discovery by transferring labels from the labelled data the unlabelled data via feature ranking statistics. The model needs to be pretrained sequentially for rotation prediction (Gidaris et al., 2018) and supervised classification. (Hsu et al., 2018a) and (Hsu et al., 2019) can also be applied to discover new categories in image datasets. Both methods require to maintain two models separately. One model is for binary label prediction and the other is for clustering. In contrast, our method is a flexible end-to-end framework which can be applied to single- and multi-modal data. As will be shown in the experiments, our method substantially outperforms others in all benchmarks.

## 3 METHOD

Given an unlabelled collection of instances $x_i^u \in D^u$, our objective is to automatically partition these instances into $C^u$ different semantic groups. We also assume that there is a labelled collection of instances $\left(x_i^l, y_i^l\right) \in D^l$ where $y_i^l \in \left\{1, \ldots, C^l\right\}$, from which we want to transfer the knowledge to the unlabelled data so that the unlabelled data can be grouped into proper classes, while the classes in $D^l$ and $D^u$ are different but relevant. Each $x_i \in D^u \cup D^l$ can be either an image or a multi-modal video consisting of a visual stream $x_i^v$ and the corresponding audio stream $x_i^a$.

Our approach is an end-to-end trainable framework that can jointly learn the representation and clustering assignment for the unlabelled data. Figure 1 shows the overview of our approach. Consider a video clip $x_i = (x_i^v, x_i^a)$, the visual encoder $f_v$ and the audio encoder $f_a$ first encode the visual and audio streams into two feature vectors $z_i^v$ and $z_i^a$, which are then concatenated to form the global representation $z_i = [z_i^v, z_i^a]$ for the input video clip. A projection function $\eta$ then project $z_i$ to $\bar{z}_i$ to fuse multi-modal feature in a compact representation space. Note that $\eta$ is an identity mapping function for single-modal data. If $x_i$ is from labelled data, $\bar{z}_i$ will be sent to the linear head $\phi^l$ for supervised learning. Otherwise, it will be sent to the linear head $\phi^u$ for new category discovery. In addition, $z_i^v$ and $z_i^a$ are further encoded to $\hat{z}_i^v$ and $\hat{z}_i^a$ by $h_v$ and $h_a$ respectively for contrastive learning to bootstrap the representation learning on both labelled and unlabelled data. $h_v$ and $h_a$ are two MLP functions followed by $\ell_2$ normalization, which is a common practice in contrastive learning.

The end-to-end training of the model consists of several important components including training on labelled data with full supervision, constrative learning with instance and category discrimination on both labelled and unlabelled data, and training on unlabelled data with pair-wise pseudo labels transferred from the representation jointly learned by the previous two. To effectively transfer knowledge from the labelled to unlabelled data, we employ the winner-take-all (WTA) hashing algorithm on $\bar{z}_i$. Next, we will introduce our extended contrastive learning for novel category discovery by considering both instance and category discrimination, and knowledge transfer via the WTA hashing in more details.

### 3.1 UNIFIED CONTRASTIVE LEARNING ON LABELLED AND UNLABELLED DATA

Given a batch of $N$ data points that are randomly drawn from both $D^u$ and $D^l$, i.e., $x_i \sim Unif(D^u \cup D^l)$ for all the elements $x_i$ in batch $B$, our objective is to extract as much information as possible from $B$ to learn a representation that can be used to cluster unknown classes. Recent developments in contrastive learning focus on learning image representation under self-supervised scenario, where no labels are available, therefore it can be thought of as contrasting instance representations from a pair of data augmentation (positive) to those of other samples (negative). In our case, we have a mixed setting with both labelled and unlabelled data. We adopt the contrastive learning to jointly learn from the mixed dataset for novel category discovery. Next, we first demonstrate our unified contrastive learning on single-modal data, and then extend it to the more flexible multi-modal scenario.

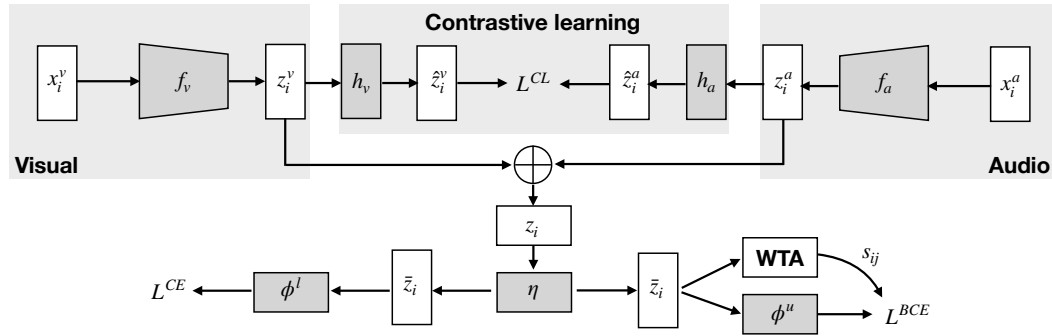

Figure 1: **Overview of our end-to-end framework**. For multi-modal videos, our framework consists of two feature encoders $f_v$ and $f_a$, two MLPs $h_v$ and $h_a$ for contrastive learning, one fusion layer $\eta$, and two linear heads $\phi^l$ and $\phi^u$ for classification and clustering. The training signal for $\phi^u$ is obtained by WTA on-the-fly. For single-modal images, the audio encoder $f_a$ is omitted and $\eta$ turns to be an identity mapping function.

### 3.1.1 SINGLE-MODAL LEARNING

Let $x_i$ be a single-modal data point and $i \in \mathcal{N} = \{1, \ldots, 2N\}$ be the index of pairs of augmented samples from the batch $B$. With the embedded representation $\hat{z}_i$ of $x_i$, we adopt NCE (Gutmann & Hyvärinen, 2010) as our contrastive loss function for instance discrimination, which can be written as:

$$\mathcal{L}_i^{NCE-I} = -\log \frac{\exp\left(\hat{z}_i \cdot \hat{z}_{i'}/\tau\right)}{\sum_n \mathbb{1}_{[n \neq i]} \exp\left(\hat{z}_i \cdot \hat{z}_n/\tau\right)}, \tag{1}$$

where $\hat{z}_{i'}$ is the augmented counterpart of $\hat{z}_i$, $\mathbb{1}_{[n \neq i]}$ is an indicator function evaluating 1 *iff* $n \neq i$, $\tau$ is a scalar temperature. This loss is widely used in conventional contrastive learning approaches.

In our case, some data points in the mini-batch $B$ are accompanied with labels. If $x_i$ is a labelled data point in the batch, besides considering only $x_i$ and its transformed counterpart as a positive pair, other data points from the same class as $x_i$ can also be paired with $x_i$ to form more positive pairs to be pulled together, allowing category discrimination. The contrastive loss for category discrimination with these additional positive pairs can be written as:

$$\mathcal{L}_i^{NCE-C} = -\frac{1}{|Q(i)|} \sum_{q \in Q(i)} \log \frac{\exp\left(\hat{z}_i \cdot \hat{z}_q/\tau\right)}{\sum_n \mathbb{1}_{[n \neq i]} \exp\left(\hat{z}_i \cdot \hat{z}_n/\tau\right)}, \tag{2}$$

where $Q(i) = \{q \in \mathcal{N} \setminus i : y_q = y_i\}$ denotes the indices of other data points which have the same label as $x_i$ in the batch $B$. Note that this is effective only for labelled data, as $Q(i) = \emptyset$ for unlabelled data. Therefore, the unified contrastive loss can be written as:

$$\mathcal{L}^{CL} = \frac{1}{2N} \sum_i^{2N} \left(\mathcal{L}_i^{NCE-I} + \mathcal{L}_i^{NCE-C}\right). \tag{3}$$

Note that there is only one positive pair in $\mathcal{L}_i^{NCE-I}$ which aims at instance discrimination, whereas multiple other samples from same class are considered as positive in $\mathcal{L}_i^{NCE-C}$ which aims at class discrimination. $\mathcal{L}_i^{NCE-I}$ and $\mathcal{L}_i^{NCE-C}$ have exactly the same denominator, which is the summation of $2N - 1$ scores for both positive and negative pairs. In this way, both labelled and unlabelled data are used for representation learning while making full use of the labels contained in the labelled data. $\mathcal{L}_i^{NCE-C}$ is a critical complementary to $\mathcal{L}_i^{NCE-I}$, resulting in a more discriminative representation space for robust clustering as will be seen in the experiments.

### 3.1.2 MULTI-MODAL LEARNING

For multi-modal data like videos, besides the conventional within-modal contrastive learning for single-modal data like images, we can also have the additional cross-modal option. As noted

by (Alwassel et al., 2019; Morgado et al., 2020), cross-modal agreement leads to better representation than within-modal agreement in self-supervised representation learning for supervised downstream tasks like object recognition and detection. However, our setting contains both labelled data and unlabelled data, resulting in a mixture of instance discrimination and class discrimination. It is not immediately obvious whether the within-modal or cross-modal choice is more effective under such a setting. For within-modal case, we can either discriminate visual or audio samples. Let the embedded representation for a multi-modal data point $x_i$ be $\hat{z}_i = \{\hat{z}_i^v, \hat{z}_i^a\}$. We define modality selecting functions $g_0$ and $g_1$ to allow either within-modal discrimination (e.g., $g_0(\hat{z}_i) = \hat{z}_i^v$ and $g_1(\hat{z}_i) = \hat{z}_i^v$) or cross-modal discrimination (e.g., $g_0(\hat{z}_i) = \hat{z}_i^v$ and $g_1(\hat{z}_i) = \hat{z}_i^a$). The instance discrimination and category discrimination objectives in Eq. (1) and Eq. (2) can then be rewritten as

$$\mathcal{L}_i^{NCE-I} = -\log \frac{\exp\left(g_0(\hat{z}_i) \cdot g_1(\hat{z}_{i'})/\tau\right)}{\sum_n \mathbb{1}_{[n \neq i]} \exp\left(g_0(\hat{z}_i) \cdot g_1(\hat{z}_n/\tau)\right)}, \tag{4}$$

and

$$\mathcal{L}_i^{NCE-C} = -\frac{1}{|Q(i)|} \sum_{q \in Q(i)} \log \frac{\exp\left(g_0(\hat{z}_i) \cdot g_1(\hat{z}_q)/\tau\right)}{\sum_n \mathbb{1}_{[n \neq i]} \exp\left(g_0(\hat{z}_i) \cdot g_1(\hat{z}_n)/\tau\right)}. \tag{5}$$

As will be shown in the experiments, after investigating different strategies, we find that the cross-modal contrastive learning can produce better representation for novel category discovery. This might be due to the fact that cross-modal representations are intrinsically different, thus reducing the chance of obtaining trivial solutions such as discriminating objects by only verifying the most salient features in visual or audio modality alone. Therefore, unless stated otherwise, the modality selecting functions $g_0$ and $g_1$ choose visual and audio representations respectively in our formulation.

## 3.2 Knowledge transfer using Winner-take-all hash

To leverage the labelled data to help novel category discovery in unlabelled data, we transfer knowledge from labelled data to unlabelled data by adopting the Winner-Take-All (WTA) hash (Yagnik et al., 2011) during training. WTA is a sparse embedding method that maps the feature vectors into integer codes. In this work, we employ WTA to measure the similarity between each pair of unlabelled data points from new categories in the shared embedding space of both labelled and unlabelled data, so that we can transfer knowledge from the labelled categories to the unlabelled ones.

The idea of WTA (Yagnik et al., 2011) is to measure similarity between high-dimensional feature vectors by comparing multiple partial ranking statistics. The WTA algorithm works as follows. First, we randomly generate a set of $H$ permutations $\mathcal{P} = \{\rho_1, ..., \rho_H\}$. For an unlabelled sample $x_i \in D^u$, we extract its feature vector $\bar{z}_i$. We then apply each $\rho_h \in \mathcal{P}$ on $\bar{z}_i$ to obtain a transformed feature vector $\rho_h(\bar{z}_i)$, i.e., a shuffled version of $\bar{z}_i$. Let $c_i^h$ be the index of maximum value in the first $k$ elements of $\rho_h(\bar{z}_i)$. We can then obtain the WTA hash code by $c_i = (c_i^1, ..., c_i^H)$ for each $x_i$.

In our case, we employ WTA hash code to measure the similarity $s_{ij}$ between $x_i$ and $x_j$ to generate pairwise pseudo labels for novel category discovery, by simply comparing their WTA hash codes $c_i$ and $c_j$:

$$s_{ij} = \begin{cases} 1, & \mathbf{1}^T \cdot (c_i = c_j) >= \mu \\ 0, & \text{otherwise} \end{cases}, \tag{6}$$

where $\mu$ is an empirical scalar threshold. Note that WTA is only applied during training to generate binary pseudo labels, and it is not needed at test time.

As discussed in (Yagnik et al., 2011), the precise values in the high-dimensional embedding is often not important, and the relative magnitude matters more. While requiring the total orderings to be identical is too strict for real application, as noise inevitably exists even using ranking statistics, therefore, WTA introduces multiple partial order statistics. Besides introducing more resilience to noise, we further emphasize that the partial orders for local rank correlation also captures the relative structural information of the objects. Intuitively, the global ranking statistics may only consider the most salient features in the embedding space, while the multiple partial orders spread more in the embedding space, thus capturing more structural information. As shown in (Zhou et al., 2020), modern CNNs are very likely to make decisions by focusing on the salient patterns while overlooking the holistic structural composition. As will be seen in the experiments, WTA outperforms other alternatives for generating pairwise pseudo labels to discover new categories.

After applying the WTA algorithm, we can obtain pairwise similarity $s_{ij}$ for each pair of unlabelled data points with Eq. (6). Assume we have $M$ unlabelled samples in a mini-batch. By using these pairwise similarities as pseudo labels, we can then train the model with binary cross-entropy loss to simultaneously learn representation and cluster assignments on the unlabelled data from new classes:

$$\mathcal{L}^{BCE} = -\frac{1}{M^2} \sum_{i=1}^{M} \sum_{j=1}^{M} \left[ s_{ij} \log \phi^u \left( \bar{z}_i \right)^\top \phi^u \left( \bar{z}_j \right) + (1 - s_{ij}) \log \left( 1 - \phi^u \left( \bar{z}_i \right)^\top \phi^u \left( \bar{z}_j \right) \right) \right], \quad (7)$$

where $\phi^u : \mathbb{R}^d \to \mathbb{R}^{C^u}$ is a non-linear function mapping $\bar{z}_i$ into an embedding space with the same dimension as the number of classes in the unlabelled data followed by softmax normalization. In this way, we can obtain the cluster assignment for each unlabelled sample by indexing the location of the maximum value in $\phi^u(\bar{z}_i)$ after training, without requiring another offline clustering procedure for class assignment.

### 3.3 JOINT LEARNING OBJECTIVE

Inspired by the literature of semi-supervised learning (Tarvainen & Valpola, 2017), we include a consistency regularization loss on both labelled and unlabelled data. The purpose of such a consistency loss is to enforce the class predictions on a data point $x_i$ and its transformed counter part $t(x_i)$ to be the same. This is especially important for unlabelled data samples. By enforcing the consistency, $x_i$ and $t(x_i)$ will be treated as a positive pair regardless of the WTA hash code, as the WTA hash code of $x_i$ might be different from that of $t(x_i)$, thus smoothing the training. The consistency loss is commonly implemented as the mean squared error (MSE) between the class predictions. Let $\mathcal{L}^{MSE}$ be the consistency loss and $\mathcal{L}^{CE}$ be the cross-entropy loss on labelled data. The consistency loss between $x_i$ and its transformed version $x_i'$ is defined as

$$\mathcal{L}_i^{MSE} = (\phi(\bar{z}_i) - \phi(\bar{z}_i'))^2, \quad (8)$$

where $\phi$ is $\phi^l$ or $\phi^u$ depending on the input. The overall training loss of our end-to-end framework can then be written as

$$\mathcal{L} = \mathcal{L}^{CE} + \mathcal{L}^{BCE} + (1 - \omega(r))\mathcal{L}^{CL} + \omega(r)\mathcal{L}^{MSE}, \quad (9)$$

where $\omega(r)$ is a ramp-up function slowly increasing from 0 to 1 along with the training. In our experiment, we follow (Laine & Aila, 2017) to use $\omega(r) = \lambda e^{-5\left(1 - \frac{r}{T}\right)^2}$ where $r$, $T$ and $\lambda$ are current epoch number, total number of epochs and a positive scalar factor. We set $(1 - \omega(r))$ as the weight for contrastive learning. At the early stages, the cluster assignment predictions are noisy and we expect the model to focus more on representation learning, thus a higher weight is set to representation learning and a lower weight is set for consistency. In the late stages, the representation is good enough and we would like the model to focus on novel category discovery, therefore the weight is higher for consistency loss and lower for contrastive learning.

## 4 EXPERIMENTS

**Benchmarks and evaluation metric.** We comprehensively evaluate our approach for novel category discovery on large-scale image benchmarks including ImageNet (Deng et al., 2009)/CIFAR-10 (Krizhevsky & Hinton, 2009)/CIFAR-100 (Krizhevsky & Hinton, 2009) and video benchmarks including Kinetics-400 (Kay et al., 2017)/ VGG-Sound (Chen et al., 2020a). We follow (Han et al., 2020) to split the ImageNet/CIFAR-10/CIFAR-100 to have 30/5/20 classes in the unlabelled classes. For fair comparison with (Han et al., 2019; 2020; Hsu et al., 2018b) three 30-class splits are used for ImageNet, and the results are averaged over the three splits. We split Kinetics-400/VGG-Sound to have 50/39 classes in the unlabelled data, which are much more challenging than the image benchmarks. As only URLs are publicly available for video datasets, by the time of our experiments, we have 170k video clips with sound in Kinetics-400 and 183k videos with sound in VGG-Sound. To measure the novel category discovery accuracy, we adopt the widely used average clustering accuracy defined as: $\max_{e \in \mathcal{P}(C^u)} = \frac{1}{U} \sum_{i=1}^{U} \mathbb{1}\{y_i^u = e(\hat{y}_i^u)\}$, where $y_i^u$ and $\hat{y}_i^u$ denote the ground-truth label and predicted cluster assignment for each unlabelled data point, $U$ is the total number of unlabelled instances in the whole dataset, $\mathcal{P}(C^u)$ denotes the set of all possible permutations of $C^u$ elements,

$e$ is an arbitrary permutation in $\mathcal{P}(C^u)$. We obtain the optimal permutation $e^*$ by Hungarian algorithm (Kuhn, 1955). Note that as no supervision is used for unlabelled data, the same data are used for both training and evaluation following standard practice (Ji et al., 2019; Han et al., 2019).

**Implementation details.** We use R3D-18 (Tran et al., 2018; Ji et al., 2012) as the video encoder and ResNet-18 (He et al., 2016) as the image and audio encoder. The feature vector dimension is 512 for both encoders. The MLPs for contrastive learning consist of a hidden layer of size 512, a linear layer of size 128, and an $\ell$-2 normalization layer (Wu et al., 2018). The output dimension of the hidden layer $\eta$ is 512. On image benchmarks, we follow SimCLR (Chen et al., 2020b) to randomly apply cropping, resizing, horizontal flip, color distortion, and Gaussian blur for data augmentation. On video benchmarks, we follow (Szegedy et al., 2015) to use the input video and audio clips of 1 and 2 second duration respectively. Video frames are resized such that the shorter side has a size of 128, followed by random cropping with size 112. We preprocess the audio by randomly sampling within 0.5 seconds of the video and compute a log mel bank features with 257 filters and 199 time-frames, followed by SpecAugment (Park et al., 2019). During evaluation, we follow (Patrick et al., 2020) to uniformly sample 10 clips from each video and take the mean score for prediction. For WTA, we set $H$ to be equal to the feature dimension (i.e., 512), and follow (Yagnik et al., 2011) to set $k$ as 4. For the threshold $\mu$, we empirically set to 260 in our experiments. We train our models with SGD (Sutskever et al., 2013) and use a batch size of 1024 for all benchmarks except CIFAR-10/CIFAR-100, which we use a batch size of 256. The models are trained with $8 \times 8$ TPUv2 Dragonfish devices.

## 4.1 ABLATION STUDY

**Component analysis.** We validate the effectiveness of each component of our method during training on both image and video benchmarks. The results are reported in table 1. As can be seen, all components are important to our method. Removing the BCE loss causes the most performance drop. Namely, the ACC drops from $93.4\% \rightarrow 32.2\%$ on CIFAR-10, $76.43\% \rightarrow 15.5\%$ on CIFAR-100 and $56.5\% \rightarrow 5.8\%$ on Kinetics-400, suggesting that the WTA-hashing comparison can indeed generate reliable pairwise pseudo labels for the BCE loss. Removing the contrastive learning leads to significant performance drop (row 4 vs row 6). By comparing rows 5 and 6, we can see that instance discrimination (NCE-I) alone is not good enough for the task of novel category discovery, and incorporating the category discrimination (NCE-C) is very effective. Please refer to the supplementary for comparison between WTA and other alternatives like cosine similarity.

Table 1: **Ablation study on image and video benchmarks.** MSE: consistency constraint; CE: cross entropy loss; BCE: binary cross entropy loss; NCE-I: NCE for instance discrimination; NCE-C: NCE for category discrimination. Results on CIFAR-10/CIFAR-100 are averaged over 10 runs.

| No | MSE | CE | BCE | NCE-I | NCE-C | CIFAR-10 | CIFAR-100 | Kinetics-400 |
|----|-----|----|----|-------|-------|----------|-----------|--------------|
| (1) | ✗ | ✓ | ✓ | ✓ | ✓ | $91.3 \pm 1.4\%$ | $74.7 \pm 2.9\%$ | $50.9\%$ |
| (2) | ✓ | ✗ | ✓ | ✓ | ✓ | $77.4 \pm 5.4\%$ | $68.5 \pm 4.7\%$ | $24.5\%$ |
| (3) | ✓ | ✓ | ✗ | ✓ | ✓ | $32.2 \pm 1.8\%$ | $15.5 \pm 1.2\%$ | $5.8\%$ |
| (4) | ✓ | ✓ | ✓ | ✗ | ✗ | $89.8 \pm 0.9\%$ | $70.1 \pm 4.5\%$ | $47.6\%$ |
| (5) | ✓ | ✓ | ✓ | ✓ | ✗ | $91.6 \pm 1.2\%$ | $75.2 \pm 3.1\%$ | $51.7\%$ |
| (6) | ✓ | ✓ | ✓ | ✓ | ✓ | $\mathbf{93.4 \pm 0.6\%}$ | $\mathbf{76.43 \pm 2.8\%}$ | $\mathbf{56.5\%}$ |

**Multi-modal contrastive learning.** As mentioned in section 3.1, for multi-modal data like videos, we can have multiple contrastive options. Namely, we can conduct within-modal and cross-modal contrastive learning for each of NCE-I and NCE-C. We present the results of different options in table 2. By comparing rows 1 and 4, we can see that cross-modal is more effective than within-modal instance discrimination. It can be seen from rows 1-3 and rows 4-6 that NCE-C can effectively improve the representation for novel category discovery. Also, it is important to consistently use either within-modal or cross-modal discrimination for NCE-I and NCE-C (row 2 vs row 3; row 5 vs row 6). Overall, cross-modal contrastive learning for NCE-I and NCE-C (row 6) performs notably better than all other cases. We hypothesize this is because enforcing the cross-modal agreement can avoid the modal falling into the trivial solution, such as simply identifying the most salient visual or audio features. The modality discrepancy can effectively avoid such cases, thus improving the quality of learned representation.

Table 2: **Multi-modal contrastive learning.** "$(a, a)$" denotes the audio-modal contrastive; "$(v, v)$" denotes visual-modal contrastive; and "$(a, v)$" cross-modal contrastive.

| No | NCE-I | NCE-C | Kinetics-400 |
|----|-------|-------|--------------|
| (1) | $(a, a) + (v, v)$ | - | 46.2% |
| (2) | $(a, a) + (v, v)$ | $(a, a) + (v, v)$ | 51.4% |
| (3) | $(a, a) + (v, v)$ | $(a, v)$ | 49.1% |
| (4) | $(a, v)$ | - | 51.7% |
| (5) | $(a, v)$ | $(a, a) + (v, v)$ | 56.1% |
| (6) | $(a, v)$ | $(a, v)$ | **56.5%** |

## 4.2 NOVEL CATEGORY DISCOVERY ON IMAGE BENCHMARKS

In table 3, we compare our approach with the $k$-means baseline and the previous state-of-the-art methods. By comparing row 10 with rows 1-6, we can clearly see that our method substantially outperforms the $k$-means baseline and existing methods. For example, our method outperforms the previous state-of-the-art method (Han et al., 2020), denoted as RS in row 5, by 3%/3.2%/4.2% on CIFAR-10/CIFAR-100/ImageNet. We also report the incremental learning result of RS (Han et al., 2020) in row 6, where they introduce an additional step to create interactions between labelled heads and unlabelled heads. The $k$-means baseline in row 1 is evaluated on the features of unlabelled data extracted using the model pretrained on labelled data by cross-entropy loss. We further enhance the $k$-means baseline, KCL (Hsu et al., 2018a) and MCL (Hsu et al., 2019) by introducing the same constrative learning used in our framework (see rows 7-9). The contrastive learning successfully boosts the performance for all of them, but the results still largely lag behind our method. For example, after the enhancement, the ACC of KCL is increased to 73.9%/57.4%/ 74.3% on CIFAR-10/CIFAR-100/ImageNet, while the results of our method are 93.4%/76.4%/86.7%, demonstrating the effectiveness of our approach on image benchmarks.

Table 3: **Novel category discovery on image benchmarks.** "CL" denotes the contrastive learning we introduce in section 3.1, which performs both instance discrimination and category discrimination.

| No | Method | CIFAR-10 | CIFAR-100 | ImageNet |
|----|--------|----------|-----------|----------|
| (1) | $k$-means (MacQueen, 1967) | $65.5 \pm 0.0\%$ | $56.6 \pm 1.6\%$ | 71.9% |
| (2) | KCL (Hsu et al., 2018a) | $66.5 \pm 3.9\%$ | $14.3 \pm 1.3\%$ | 73.8% |
| (3) | MCL (Hsu et al., 2019) | $64.2 \pm 0.1\%$ | $21.3 \pm 3.4\%$ | 74.4% |
| (4) | DTC (Han et al., 2019) w/ S.S. | $88.7 \pm 0.3\%$ | $67.3 \pm 1.2\%$ | - |
| (5) | RS (Han et al., 2020) | $90.4 \pm 0.5\%$ | $73.2 \pm 2.1\%$ | 82.5% |
| (6) | RS (Han et al., 2020) w/ I.L. | $91.7 \pm 0.9\%$ | $75.2 \pm 4.2\%$ | - |
| (7) | $k$-means (MacQueen, 1967) w/ CL | $73.2 \pm 0.1\%$ | $58.8 \pm 1.5\%$ | 73.1% |
| (8) | KCL (Hsu et al., 2018a) w/ CL | $73.9 \pm 0.1\%$ | $57.4 \pm 3.6\%$ | 74.3% |
| (9) | MCL (Hsu et al., 2019) w/ CL | $72.3 \pm 0.2\%$ | $50.8 \pm 3.1\%$ | 75.2% |
| (10) | Ours | $\mathbf{93.4 \pm 0.6\%}$ | $\mathbf{76.4 \pm 2.8\%}$ | **86.7%** |

## 4.3 NOVEL CATEGORY DISCOVERY ON VIDEO BENCHMARKS

In table 4, we compare our framework with $k$-means baseline and (Han et al., 2020) on Kinetics-400 and VGG-Sound benchmarks. In rows 1-3, we train the models on labelled data with cross-entropy loss, and extract the features on unlabelled data for $k$-means. We validate features from multi-modalities as well of each single-modality. The multi-modal features are more effective than the single-modal features. To compare with the previous state-of-the-art method (Han et al., 2020), which is designed for image category discovery, we implement its multi-modal counterpart (row 4) and also enhance it by replacing the original RotNet pretraining with our improved contrastive learning (row 5). It can be see that the contrastive learning introduced in our framework can further boost the performance of (Han et al., 2020), and our approach consistently outperforms the $k$-means baseline and (Han et al., 2020) by a large margin. For example, our method achieves 56.5%/50.0% ACC on Kinetics-400/VGG-Sound, while (Han et al., 2020) only gives 31.2%/38.6% ACC.

Please refer to the appendix for more results.

Table 4: **Novel category discovery on video benchmarks.**

| No | Method | audio | video | Kinetics-400 | VGG-Sound |
|----|--------|-------|-------|--------------|-----------|
| (1) | $k$-means (MacQueen, 1967) | ✗ | ✓ | 40.9% | 32.4% |
| (2) | $k$-means (MacQueen, 1967) | ✓ | ✗ | 18.7% | 41.7% |
| (3) | $k$-means (MacQueen, 1967) | ✓ | ✓ | 41.1% | 43.4% |
| (4) | RS (Han et al., 2020) w/ RotNet | ✓ | ✓ | 31.2% | 38.6% |
| (5) | RS (Han et al., 2020) w/ CL | ✓ | ✓ | 33.5% | 42.2% |
| (6) | Ours | ✓ | ✓ | **56.5%** | **50.0%** |

## 5 CONCLUSION

In this paper, we have presented a flexible end-to-end framework to tackle the challenging problem of novel category discovery. First, we extended the conventional contrastive learning to perform instance discrimination as well as category discrimination jointly by making full use of the labelled data and unlabelled data. In this way, the learned representation is not biased towards labelled or unlabelled data, thus can be effectively used to discover new categories. Second, to successfully transfer knowledge from the labelled data to the unlabelled data, we employed the WTA hashing algorithm to generate pair-wise weak pseudo labels for training on unlabelled data, which is the key to automatically partition the unlabelled data into proper groups after training. Third, for multi-modal data, we investigated different ways of contrastive learning and we empirically found that cross-modal noise contrastive estimation performs consistently better than other options. Last, we thoroughly evaluated our approach on challenging image and video benchmarks and obtain superior results in all cases, demonstrating the effectiveness of our approach.

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

## A    REPLACE STATISTICAL RANKING WITH WTA IN RS

To better understand the role of WTA in our framework, we investigate RS Han et al. (2020) furhter by replacing the ranking statistics with the WTA used in our framework. The results are reported in Table 5. It can bee seen that simply replacing ranking statistics with WTA cannot boost the performance. Other components of our approach are also important, as shown in Table 1. Integrating into a unified framework, it achieves superior performance.

Table 5: **Replacing ranking statistics in Han et al. (2020) by WTA**

| Dataset | RS Han et al. (2020) | RS w/ WTA |
|---|---|---|
| CIFAR-10 | 90.4 | 90.1 |
| CIFAR-100 | 73.2 | 67.3 |

## B    PARAMETER ANALYSIS FOR WTA

We obtain the two hyperparamters of WTA, i.e. threshold $\mu$ and window size $k$, by examining different values on the labelled data. We further split the labelled data into a smaller labelled subset and an unlabeled subset (i.e. pretending part of the labelled data to be unlabelled), and find $\mu$ and $k$ that give the best results on the unlabelled subset. More specifically, for Kinetics-400, we split 350-class labelled set into a 300-class labelled subset and a 50-class unlabelled subset. For VGG-Sound, we split the 270-class labelled subset into 231-class labelled subset and 39-class unlabelled subset respectively. The number of unlabelled classes used here are the same as the unlabelled classes in the main experiments.

Taking the empirical WTA window size $k = 4$ in Yagnik et al. (2011), we experiment with different threshold $\mu$ and report the results in table 6. We find that the performance is generally stable for $\mu$ greater than 200. We set $\mu$ to 240 in our experiments. Note that the feature vectors have a dimension of 512.

Table 6: **Performance of different WTA threshold on video benchmarks.**

| Dataset | 130 | 180 | 200 | 240 | 260 | 300 |
|---|---|---|---|---|---|---|
| Kinetcis-400 | 19.4 | 39.7 | 55.2 | 54.5 | 54.8 | 54.5 |
| VGG-Sound | 21.5 | 41.6 | 49.6 | 51.3 | 50.9 | 47.9 |

Given $\mu = 240$, we sweep different $k$ and report the results in table 7. We find that the $k = 4$ and $k = 8$ perform comparably well, and they are both better than $k = 2$ and $k = 16$. Hence, we simply use the $k = 4$ in our experiments.

Table 7: **Performance of different WTA window size on video benchmarks.**

| Dataset | $k = 2$ | $k = 4$ | $k = 8$ | $k = 16$ |
|---|---|---|---|---|
| Kinetcis-400 | 52.6 | 56.7 | 55.9 | 49.4 |
| VGG-Sound | 46.2 | 51.8 | 51.0 | 48.3 |

## C    COMPARING WTA WITH OTHER ALTERNATIVES

We compare WTA and other alternatives to transfer pseudo labels on-the-fly in our framework. We compare WTA with cosine similarity, ranking statistics (Han et al., 2020), and nearest-neighbour. Table 8 shows the results. We can see that WTA performs significantly better than all other alternatives.

Table 8: **WTA vs other alternatives.**

| Method | Kinetics-400 | VGG-Sound |
|---|---|---|
| cosine | 35.4% | 29.8% |
| nearest-neighbour | 19.6% | 11.5% |
| ranking statistics (Han et al., 2020) | 37.4% | 38.6% |
| WTA | 56.5% | 50.0% |

## D  UNKNOWN CLASS NUMBER IN UNLABELLED DATA

Following Han et al. (2020), we assume the number of the classes, $C^u$, in the unlabelled data is known a-priori. When $C^u$ is not known, we can use the method introduced in DTC Han et al. (2019) to estimate $C^u$ first, and then substitute the estimated number into our framework. We evaluate the performance of our approach on ImageNet using the unknown category numbers estimated by DTC. The estimates are 34/32/31 and the ground-truth numbers are 30/30/30 on the three unlabelled subsets. The average accuracy over three subsets is $84.1\%$ which outperforms Han et al. (2020) by $3.6\%$.

## E  UNSUPERVISED CLUSTERING

We further experiment with our approach for pure unsupervised clustering on the unlabelled subset of CIFAR10 and CIFAR100, which contains 5 and 20 classes respectively, by simply dropping the labelled data. Our method achieves $84.6\%$ and $61.5\%$ on the two datasets respectively, while the results by $k$-means baseline (using features extracted by the model trained on the labelled subset) are $65.5\%$ and $56.5\%$ respectively (see table 3), showing the superiority of our approach. This reveals that our method is also an effective clustering method.

## F  QUALITATIVE RESULTS

We show the qualitative results on images and videos. In fig. 2 and fig. 3 we show the t-SNE projection for the features of data from the 5 unlabelled classes in CIFAR-10. The features are extracted using the model pretrained on the labelled data ( fig. 2) and using our model ( fig. 3) respectively. We can see that the embedding is rather cluttered using the model pretrained on the labelled data, while our model can successfully partition the unlabelled data into tight semantic groups. Similarly, we compare the features on videos in fig. 4 and fig. 5. For visualisation purpose, we randomly choose unlabelled instances from 10 classes from Kinetics-400, we can see that our model can successfully separate novel classes into compact groups, while the novel categories are projected very close to each other for the baseline model trained with full supervision on the labelled data.

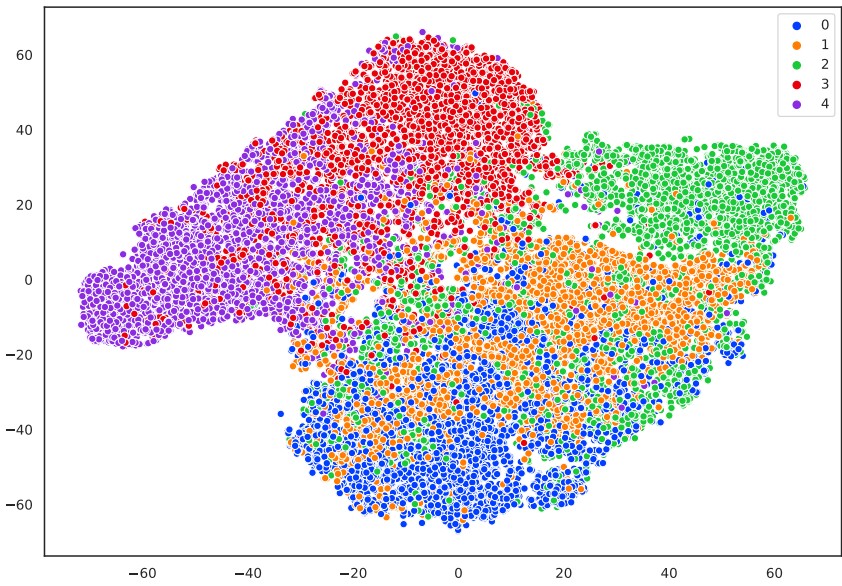

Figure 2: **t-SNE visualization of features from unlabelled data on CIFAR-10, using the model pretrained on the labelled data with fully supervised training.**

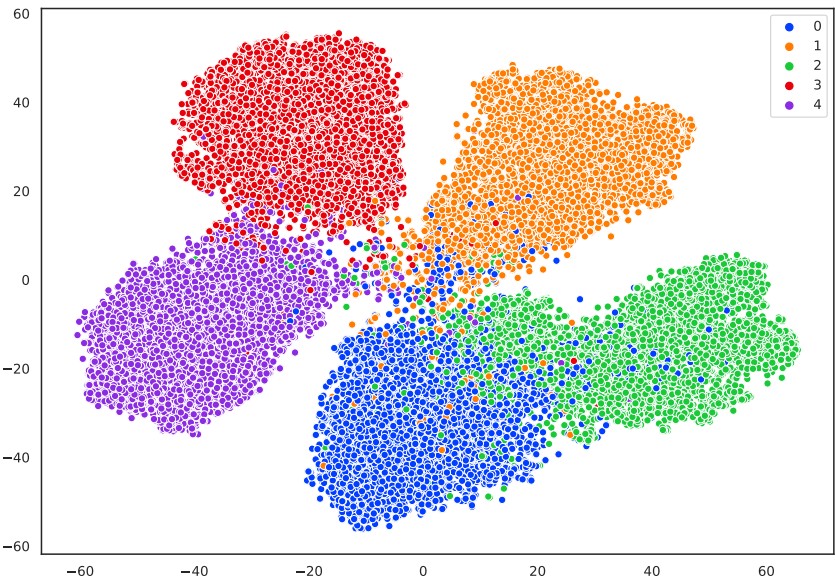

Figure 3: **t-SNE visualization of features from unlabelled data on CIFAR-10, using our model after end-to-end training on both labelled and unlabelled data.**

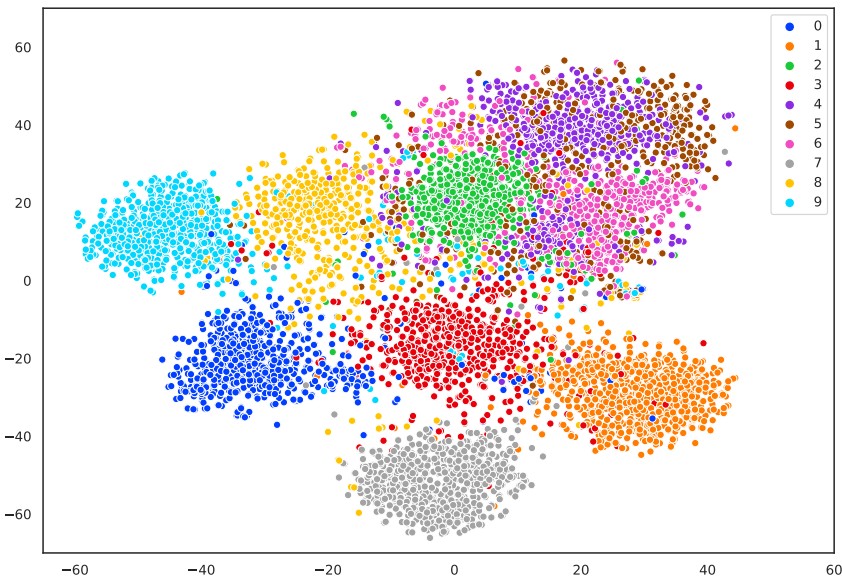

Figure 4: **t-SNE visualization of features from unlabelled data on Kinetics-400, using the model pretrained on the labelled data with fully supervised training.**

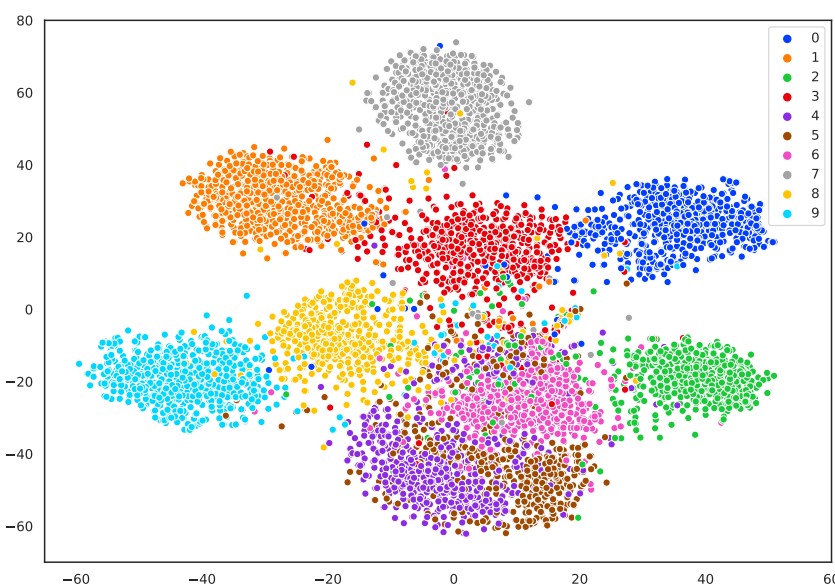

Figure 5: **t-SNE visualization of features from unlabelled data on Kinetics-400, using our model after end-to-end training on both labelled and unlabelled data.**

