# OpenReview forum: "A Flexible Framework for Discovering Novel Categories with Contrastive Learning"
_ICLR.cc/2021/Conference — Reject_

### Official Review · AnonReviewer1 · 2020-10-26
**Official Blind Review #1**

**Rating:** 5
**Confidence:** 3

**Review:**

The paper proposes a new framework for novel category discovery, i.e., assigning novel class labels of an unlabeled dataset from a given (different but relevant) labeled dataset. Specifically, the framework (a) combines SimCLR, supervised contrastive learning, supervised learning, and consistency loss to learn a better representation from a mixed dataset of labeled and unlabeled samples, and (b) proposes a loss for learning a cluster assignment based on the winner-take-all coding of the learned representation from (a). Experimental results on image and video datasets confirm the effectiveness of the proposed framework compared to prior works, in terms of the average clustering accuracy on samples with unseen classes.

Overall, the paper is well-written. The method is clearly explained given the complexity of the method itself. I liked their strong experimental results, and a thorough ablation study performed to verify the individual effects of each component of the method. It is also inspiring to see their another ablation study to explore different modality pairs possible in multi-modal contrastive learning, which concludes that contrasting against different modalities leads a better representation (at least) for the target task.
- p5, "Concretely, we randomly generate a set of H permutations ...": It would be nice if the paper could clarify what each permutation actually permutes, i.e., the domain. Also, are those permutations kept unchanged for both training and inference, or else?
- Eq. 6: I feel L^CE is somewhat less motivated than the others - I found (possibly) L^NCE-C already forces the representation to be class-discriminative, then why should L^CE be crucial for the method to work, as shown in the ablation study? I think more rationale behind this could be provided in the paper.
- Eq. 6: Should L^MSE and L^CL be always jointly ramped up and down? What happens if L^MSE is not ramped-up, i.e., not weighted by w(r)?
- Could the proposed framework still work when there is no available labeled sample, i.e., for pure unsupervised clustering?
- Table 3: Why DTC+CL and RS+CL are not considered?

---
Post-rebuttal update

I have read other reviews and responses, and I am willing to decrease my score to 5.
My initial review was based on assumption that RS (Hen et al., 2020) is significantly different to this paper, but after reading other reviews and responses now I can see that it is indeed questionable. Also, It was somewhat surprising to me that the paper copied wrong values from the RS paper, as pointed out by R5, considering that RS is the closest related work of the paper.

Although I still think the proposed method is reasonable and the empirical results are impressive, I agree with other reviewers that the paper should have done more analysis on why the proposed method works to convince the reviewers (and the future readers), and to give more insights on why the combination of losses (and the use of WTA) is a right way to go.

---

> ### Author Response · Authors · 2020-11-21
> **Answer to Reviewer 1**
>
> We would like to thank the reviewer for the comments and helpful suggestions.
>
> > Q: p5, "Concretely, we randomly generate a set of H permutations ...": It would be nice if the paper could clarify what each permutation actually permutes, i.e., the domain. Also, are those permutations kept unchanged for both training and inference, or else?
>
> > A: We follow the original WTA paper to generate the permutations, which are the permutations of the indices of the values in the feature vector. The indices range from 1 to the dimension of the feature vector. Each permutation captures the salient features of a local region. The permutations are randomly generated on-the-fly during training to get the WTA code for learning on unlabelled data. During inference, WTA code is not needed, as the clustering assignment is predicted by argmax on the logits from the clustering head. We have further clarified this in Section 3.2 of the revised paper.
>
> > Q: Eq. 6: I feel L^CE is somewhat less motivated than the others - I found (possibly) L^NCE-C already forces the representation to be class-discriminative, then why should L^CE be crucial for the method to work, as shown in the ablation study? I think more rationale behind this could be provided in the paper.
>
> > A: According to our comparison row 2 vs row 4 in table 1, it seems that L^NCE-C cannot fully replace L^CE. The two losses have different purposes, the former focus on representation learning, which also enforces the uniformity of induced distribution on the hypersphere as in paper [1],  while the later focus on classification, which seems to have no intention for uniformity as in the contrastive learning. L^NCE-C is applied on the embedding space projected by the MLP, while L^CE is applied on the feature vectors.
>
> > Q: Eq. 6: Should L^MSE and L^CL be always jointly ramped up and down? What happens if L^MSE is not ramped-up, i.e., not weighted by w(r)?
>
> > A: The ramp-up plays an important role in the semi-supervised learning (SSL) literature, and we inherited the idea of ramp-up from SSL. If we do not perform ramp-up, i.e., setting w(t) as 1 for L^MSE. At the early training stage, the predictions are not reliable, thus enforcing the consistency between those unreliable predictions will prevent the model from learning better predictions.  As the learning progresses, the representation becomes better and better.  We would then like the model to focus more on novel category discovery. Thus, the model penalizes more for  the prediction inconsistency between different augmentations of the same example, and penalizes less for the representation learning. To validate this, we conducted experiments on CIFAR10 and CIFAR100 by setting w(t) as 1. The performance dropped 1.8% and 2.4% respectively.
>
> > Q: Could the proposed framework still work when there is no available labeled sample, i.e., for pure unsupervised clustering?
>
> > A: Thanks for this suggestion. We verified our approach on pure unsupervised clustering by removing the requirement for the labelled data (thus the labelled head in our framework). Our method achieves promising results without using any labels. Our fully unsupervised method achieves 84.6% and 61.5% on CIFAR10 and CIFAR100, while the results by k-means baseline (using features extracted by the model trained on the labelled subset) are 65.5% and 56.5% respectively, showing the superiority of our approach. This reveals that our method is also an effective clustering method. We have included this result in appendix E.
>
> > Q: Table 3: Why DTC+CL and RS+CL are not considered?
>
> > A: DTC [2] is significantly different from RS [3] and Ours, as it requires injecting PCA projection layers and constructing running learning targets with an exponential moving average model, which are not applicable for our case, and it requires multiple training stages. Hence we didn't include DTC + CL. Besides, as shown in the comparison in RS [3] paper, the performance of DTC [2] largely lags behind RS. Hence, we mainly compare with RS [3]. We included the RS+CL experiments in Table 4 on the video datasets, which is more challenging than images. Our method performs significantly better than RS+CL.
>
> ===
>
> [1] Wang et al. Understanding Contrastive Representation Learning through Alignment and Uniformity On the Hypersphere .
>
> [2] Han et al. Learning to discover novel visual categories via deep transfer clustering, ICCV 2019
>
> [3] Han et al. Automatically discovering and learning new visual categories with ranking statistics, ICLR 2020

---

### Official Review · AnonReviewer2 · 2020-10-28
**Good performance, but many rooms to be improved**

**Rating:** 5
**Confidence:** 4

**Review:**

Summary:

The paper proposes a contrastive learning method for novel category discovery. The authors propose loss functions that can utilize labeled data of known categories and unlabelled data like semi-supervised learning, whereby they leverage both category discrimination and instance discrimination. They also extend cross-modal discrimination for multi-modal data. But these are the side loss functions. The key component to discover novel classes is a clustering loss, for which the authors propose to use Winner-Take-All hashing to generate pseudo labels of cluster partitions.

The proposed method is evaluated on CIFAR-10/-100 and ImageNet for single modality, and Kinetics-400 and VGG-Sound for audiovisual modality, and shows notable performance improvement over the state-of-the-art.


Reasons for score:

The proposed method shows significant performance improvement for the target task, but the manuscript seems not ready for publication to this reviewer in that: 1) it is unclear specifically what makes the proposed method better than the competing method [Han et al. 2020], 2) the motivation of the proposed method sounds hand-wavy (in particular, why should the multi-modal extension be considered?), and 3) there is room to improve the paper organization further (it comes and goes).
This reviewer would like to ask for clear and logical responses to the comments.

Pros:
- Noticeable performance gain over the state-of-the-art
- The authors demonstrate that WTA hashing based clustering is effective for novel category discovery.
- Reasonable ablation study

Cons:
- Lack of motivation of the designs (in particular, multi-modal)
- The paper is written in a confusing way to identify the most significant contribution (I deem it is WTA hashing based clustering).
- The experiments are not designed to clearly reveal a key contribution.
- There is room to improve the paper organization further.


Detail comments and questions:
- After going over all the experiment including the experiments in the supplmentary, this reviewer thinks that WTA hashing based clustering is the source of the most significant gain the authors obtain. Is this a correct parse? If yes, then both the description and experiments are not presented to effectively emphasize this part. In particular, the experiment could be designed to show more clearly about this point.
	- Specifically, Tables 3 and 4 show that, even with contrastive learning (CL), the competing methods do not show notable performance gain. Then, the key difference that leads to the performance gain may come from the WTA hashing based supervision. It would have been effective to present the combination of RS [Han et al., 2020] + WTA hashing based clustering, whereby the source of the performance gain becomes clearer.
	- The ablation study in Table 1 does not replace the above suggestion, because it is obvious to have detrimental performance when omitting the closest loss to the target task (the novel class discovery) which is the WTA hashing based clustering loss. Thus, to show the effectiveness of WTA hashing based clustering in the target task regime, it should have been compared with its counterparts from the state-of-the-arts.

- Why is multi-modal important or effective for novel category discovery? It is barely discussed.

- The paper structure and organization should be improved.
	- It would have been clearer to put a separate subsection for describing the multi-modal extension case from Sec. 3.1, because the current description is written in a back and forth way. The current description would make the readers distracted to concentrate on the cores.
	- The main and key experiments should come first, and the other side experiments including the ablation study go later. Thereby, the reader can get a clear goal and achievement of the development first before going to the details.
	- The current two paragraphs right above Eq. (4) are wordy and hard to take the true intention of the context. For example, in the second paragraph, $k$ is mentioned but the discussion is already made before introducing the algorithm in the previous paragraph; thus, not smoothly linked well what the authors want to convey in the first paragraph. It would be improved by rewriting those paragraphs to clearly link between the algorithm and the design principles.

---

> ### Author Response · Authors · 2020-11-21
> **Answer to Reviewer 2 [Part 1]**
>
> We would like to thank the reviewer for the comments and helpful suggestions.
>
> > Q:  it is unclear specifically what makes the proposed method better than the competing method [Han et al. 2020] (and another comment, "After going over all the experiment including the experiments in the supplmentary, this reviewer thinks that WTA hashing based clustering is the source of the most significant gain the authors obtain. Is this a correct parse? If yes, then both the description and experiments are not presented to effectively emphasize this part. In particular, the experiment could be designed to show more clearly about this point." )
>
> > A: Our method introduced several innovations to achieve the superior performance. (1) We extend the contrastive learning for novel category discovery by jointly considering category- and instance-level discrimination, resulting in more reliable representation learning. Indeed, unsupervised contrastive learning alone has been shown to be more effective than the rotation prediction pretext task in MoCo and SimCLR papers. (2) We employ WTA to generate pairwise pseudo labels for training on the unlabelled data, which is more reliable than the ranking statistics used in RS [1]. (3) Our framework is end-to-end trainable and can deal with multi-modal data effectively. As can be seen in Table 4 and Table 5, simply replacing individual components of our approach results in sub-optimal results for RS, which shows that different components are complementary in our framework. Meanwhile, from the ablation study in Table 1, we can see that all components of our approach are important in achieving superior performance.
>
> > Q: Specifically, Tables 3 and 4 show that, even with contrastive learning (CL), the competing methods do not show notable performance gain. Then, the key difference that leads to the performance gain may come from the WTA hashing based supervision. It would have been effective to present the combination of RS [Han et al., 2020] + WTA hashing based clustering, whereby the source of the performance gain becomes clearer.
>
> > A: We follow the suggestion to validate RS [Han et al., 2020] + WTA combination, and the results are included in appendix A, Table 5. As can be seen, simply replacing the topk ranks by WTA does not lead to improvement. On the contrary, we observe performance drop, which reveals that other components of our approach is also important in improving the performance, and different components complement each other. The end-to-end training framework could better transfer knowledge from labelled to unlabelled data, and prevent the learned representation biased towards one of them compared to multiple stages sub-optimization in RS.
>
> > Q: The ablation study in Table 1 does not replace the above suggestion, because it is obvious to have detrimental performance when omitting the closest loss to the target task (the novel class discovery) which is the WTA hashing based clustering loss. Thus, to show the effectiveness of WTA hashing based clustering in the target task regime, it should have been compared with its counterparts from the state-of-the-arts.
>
> > A: We have experimented with other alternatives of WTA in Table 8 in the Appendix. As can be seen, incorporating WTA in our framework appears more effective than other methods.
>
> > Q: Why is multi-modal important or effective for novel category discovery? It is barely discussed.
>
> > A: In the context of novel category discovery, the possibility of discovering new classes with multi-modal data remains unexplored, although learning from multi-modal data has attracted a lot of attention motivated by the fact that humans naturally learn from multi-modal information. Firstly, we design the ablation study in table 2 to answer the question how to use multi-modal data for contrastive learning, we find cross-modal is more effective than within-modal. We hypothesis this is because the within-modal could fall into a trivial solution by low-level feature, e.g., within the visual stream, the color of cloth a person wears rather than the action a person conducts could trick a contrastive learning task.  Secondly, using multi-modal (visual, audio) rather than single-modal (visual) demonstrates better performance for novel category discovery, based on k-means baseline experiments in table 4 and other works in action recognition literature [2].  We added more analysis in Sec 4.1 and more discussion in the related work Sec 2. We also highlighted  the multi-modal learning in Sec 3.1.2
>
> > Q: It would have been clearer to put a separate subsection for describing the multi-modal extension case from Sec. 3.1, because the current description is written in a back and forth way. The current description would make the readers distracted to concentrate on the cores.
>
> > A: Thanks. We have revised the paper following the suggestion. In the updated version, we first introduce the single-modal case, and then generalize it to the multi-modal case.

---

> > ### Author Response · Authors · 2020-11-21
> > **Answer to Reviewer 2 [Part 2]**
> >
> > > Q: The main and key experiments should come first, and the other side experiments including the ablation study go later. Thereby, the reader can get a clear goal and achievement of the development first before going to the details.
> >
> > > A: Thanks. We agree that it is a good idea to put ablation study in the end, and would like to move the ablation to the last. However, moving it now will lead to table ID changes, which may confuse other reviewers for the tables they were referring to. We will update this in the final version of the paper.
> >
> > > Q: The current two paragraphs right above Eq. (4) are wordy and hard to take the true intention of the context. For example, in the second paragraph, $k$ is mentioned but the discussion is already made before introducing the algorithm in the previous paragraph; thus, not smoothly linked well what the authors want to convey in the first paragraph. It would be improved by rewriting those paragraphs to clearly link between the algorithm and the design principles.
> >
> > > A: Thanks. We have revised Section 3.2 following the suggestion.
> >
> > ====
> >
> > [1] Han et al. Automatically discovering and learning new visual categories with ranking statistics.
> >
> > [2] Bian et al. Revisiting the Effectiveness of Off-the-shelf Temporal Modeling Approaches for Large-scale Video Classification

---

### Official Review · AnonReviewer4 · 2020-10-29

**Rating:** 4
**Confidence:** 4

**Review:**

[Summary]
This paper addresses the problem of clustering unseen classes with the help of labeled data from an exclusive set of classes. It proposes an end-to-end learning strategy that combines a set of optimization objectives for training a deep neural network that can output a cluster index for the unlabeled data. The objectives include modified BCE for clustering, instance-wise contrastive loss, class-wise contrastive loss, consistency loss for augmented data pairs, and standard cross-entropy for the labeled data. The objectives utilize labeled and unlabeled data jointly. Extensive experiments on multiple image datasets show its advantage over SOTA in the class-discovery benchmark. The extended setup with multi-modality data demonstrates the capability of gaining additional performance when richer information is available.

[Strengths]
The experiments have good coverage on four popular datasets. The ablation of each component clearly points out the import factors to achieve good performance. The results show a significant advantage over the SOTA method.

[Weaknesses]
It is unclear what the novelty is in the proposed method compared to (Han et al. 2020). The primary difference is replacing the self-supervised learning method (rotation) with contrastive learning. The remaining learning objectives are highly similar to (Han et al. 2020). Although this paper optimizes all objectives jointly, it is unclear what the necessary change is to make this possible (or not possible in (Han et al. 2020)). Lastly, the winner-take-all-hash in the L_BCE is highly similar to the rank statistics (Han et al. 2020). The author should give an in-depth comparison against  (Han et al. 2020) and discuss what issues are mitigated by the proposed changes.

[Questions]
1. The performance of DTC and RS in Table 3 is significantly lower than the original paper. Why?
2. How will the choice of mu in eq-4 influence the performance?

-------------------------------------------
POST REBUTTAL

Thanks for the author's swift responses. I appreciate that. However, the paper still needs to provide more evidence and precise comparisons to clarify the mentioned concerns. Three major concerns are listed below:
1. Why does the proposed pipeline perform better than RS? The paper should add the proposed modifications one-by-one into the RS method and show/discuss how these individual design choices affect the performance.
2. WTA is worse than rank statistics in Appendix Table 5. Is it a counterexample of WTA? It needs more experimental supports to show WTA is a better method.  The hyper-parameters used in all experiments should be provided in a table and explain how the parameters are selected. Please make sure the same tuning budget is given to all the methods for a fair comparison.
3. What are the necessary changes to make end-to-end training possible (or not possible in RS)? The paper should summarize RS's design and explain why it can not do end-to-end training, then explain what modifications are made to make this possible. This comparison is better to be illustrated in figures.

Overall, this is an interesting paper, and the community will be benefit from it if well presented. Please consider revising and submitting to another venue.

---

> ### Author Response · Authors · 2020-11-21
> **Answer to Reviewer 4**
>
> We would like to thank the reviewer for the comments and helpful suggestions.
>
> > Q:  It is unclear what the novelty is in the proposed method compared to (Han et al. 2020). The primary difference is replacing the self-supervised learning method (rotation) with contrastive learning. The remaining learning objectives are highly similar to (Han et al. 2020). Although this paper optimizes all objectives jointly, it is unclear what the necessary change is to make this possible (or not possible in (Han et al. 2020)). Lastly, the winner-take-all-hash in the L_BCE is highly similar to the rank statistics (Han et al. 2020). The author should give an in-depth comparison against (Han et al. 2020) and discuss what issues are mitigated by the proposed changes.
>
> > A: We revised papers following the suggestions to highlight our contributions. To sum up, our approach differs from RS [2] in many important aspects.
> (1) Our framework is end-to-end trainable, while RS requires three training stages. This design avoids our model to be biased towards labelled data throughout the whole training process while effectively transferring knowledge from the labelled data to the unlabelled data.
> (2) Our model is a generic framework that can deal with single- or multi-modal data. RS(Han et al. 2020) is not applicable to multi-modal data, as all the components are designed for images, while our framework can deal with multi-modal data, and effectively transfer knowledge from labelled multi-modal data to the unlabelled one. We applied the RS [2] on multi-modal video datasets, and its performance largely lags behind our framework (see Table 4).
> (3) We extend to contrastive learning to handle both category- and instance-level discrimination to learn robust representation for novel category discovery. RS [2] adopts the rotation prediction pretext-task, which is only applicable to natural images and is mainly able to capture lower level features.
> (4) We employ WTA in our framework for robust pairwise pseudo label generation for training on unlabelled data. The ranking statistics in RS [2] considers only the topk elements in the feature representation, while WTA considers multiple partial orderings, allowing the model to take the holistic structure of the object into consideration during comparison. Meanwhile topk struggles on multi-modal data. We hypothesis this is because the features of videos are much more complicated than the images, thus simply verifying the most salient regions in the feature vectors is not robust enough to compare new classes in the videos. Further, we experimented on RS [2] by only replacing the ranking statistics with WTA, and didn’t observe performance gain. We added the results in appendix A. The results show that the other components of our approach are also important (see Table 1).
>
> > Q: The performance of DTC and RS in Table 3 is significantly lower than the original paper. Why?
>
> > A: Thanks for pointing this out. For the results of DTC [2] we took the raw results in DTC [2] paper without self-supervision. For the RS [1] results, we carelessly copied the wrong line of results in Table 1 of RS [1] paper. We have corrected this in the revised paper, and the conclusion remains unchanged: our method notably outperforms DTC and RS in all three datasets in Table 3.
>
> > Q: How will the choice of mu in eq-4 influence the performance?
>
> > A: We report the effects of different \mu in Table 6 in the appendix. In practice, we determine \mu by splitting the labelled data into two subsets: one with labels, and the other without labels. We vary different \mu and find the \mu that gives the best results on the unlabelled subset. This process is to mimic the validation process of supervised classification.
>
> ====
>
> [1]  Han et al. Automatically discovering and learning new visual categories with ranking statistics. ICLR 2020
>
> [2]  Han et al. Learning to discover novel visual categories via deep transfer clustering, ICCV 2019

---

> > ### Comment · AnonReviewer4 · 2020-11-24
> > **Question about WTA**
> >
> > If WTA is helpful, why can it not improve RS? How are the hyperparameters tuned in Appendix Table 5?

---

> > > ### Author Response · Authors · 2020-11-24
> > > **Answer for WTA**
> > >
> > > Thanks for the reviewer's quick feedback. We used the same hyperparameters as in our main experiments (the hyperparameters tuning is described in Appendix B) for WTA. For the raw RS parts, we used the hyperparameters in the original paper. We might be able to see some further improvement if we heavily tune the hyperparameters. The point we would like to highlight in Table 5 is that WTA is not the key reason for all the improvements in novel category discovery in images, and other factors are also important, e.g.,  an end-to-end training framework that leads to unbiased representation learning; instance- and category-level contrastive learning enhance the representation. Combining all these into a unified framework, our approach achieves better performance (in Table 1-3). But in multi-modal setting, we do find WTA has shown significantly gain than topk ranks (i.e., ranking statistics in RS). We hypothesize this is because the features of videos are much more complicated than that of images, thus simply verifying the most salient regions in the feature vectors is not robust enough to compare new classes in the videos.  As this setting is still at its infancy, more exploration around finding better methods to effectively learn unbiased representation, and better methods to transfer knowledge from labelled data to unlabelled data are still needed. We’d like to release the code of the paper and hope this work could be helpful to facilitate the future research for novel category discovery.

---

> > > > ### Comment · AnonReviewer4 · 2020-11-24
> > > > **hyperparameters**
> > > >
> > > > Do the same set of hyperparameters used in Table 3 for your method? Appendix Table 5 looks like a counterexample, showing that WTA is not necessarily better. It is hard to exclude the case that WTA just provides more hyperparameters for fitting the evaluated dataset.

---

> > > > > ### Author Response · Authors · 2020-11-24
> > > > > **Answer for hyperparameters**
> > > > >
> > > > > Thanks for reviewer's quick feedback.
> > > > >
> > > > > We used the same set of WTA parameters for different evaluated datasets, and also the hyper parameters not fine-tuned on the evaluated classes splits,  which might be helpful to relieve the concern of overfitting? WTA might not necessarily be better than Topk in image recognition tasks, but it is significantly better than Topk in multi-modal settings. Moreover, we have not tuned WTA upon RS framework.

---

### Official Review · AnonReviewer3 · 2020-11-01
**The framework shows impressive empirical results but the design is ad hoc and has limited novelty in some components.**

**Rating:** 6
**Confidence:** 4

**Review:**

This paper proposes an end-to-end framework for new class discovery. There are several key steps in the framework. (1)First is that they use separate losses, for instance, and class discrimination, so that even the unlabeled data can be trained jointly. (2)The second one is they use WTA hashing other than clustering algorithms to assign pseudo labels for unlabelled data. (3)The last one is they extend the framework to multi-modality settings.

Pros:
+ The paper is easy to follow. The authors have a very clear structure for the motivations, problems with the current approaches, and contributions.
+ The idea to generalize/unified instance and class discrimination in this semi-supervised setting are pretty simple and effective.
+ The final experimental results are impressive compared to previous approaches. Especially adding multi-modality and video in evaluation is appreciated.

Cons:
- The presentation of contribution two is not convincing in two ways. First, at the algorithm level, if WTA is not the paper’s contribution, I recommend authors either weaken the way they describe it in abstract/intro and shorten sec 3.2; or explain in details why not other LSH variants or even learning to hash variants for unsupervised/semi-supervised setting? The reasoning in 3.2 is mostly adapted from the original WTA paper and also the follow-up best paper in CVPR[1]. Second, in experiments, I can see when changing from WTA to Simhash, the performance drops sharply (even worse than the baseline). Does it mean that WTA is the real component that makes the framework working? I can also infer it from table 1 that when BCE is omitted, the performance is affected most.
- The framework is ad hoc. There are many losses and design options making the practicality of the framework doubtful.

Minor comments that will not affect scores:
- In the experiments section, I recommend first presenting the final results first and then ablations.

[1] Dean, Thomas, et al. "Fast, accurate detection of 100,000 object classes on a single machine."  CVPR2013

===============================================================================

After reading the rebuttals and comments from other reviewers, I decided to keep my original score.

My biggest concern hasn't been addressed. I understand there is a comparison between different alternatives in appendix A, but only WTA is outperforming RS in the existing work by a large margin. All the other reviewers and I are curious why this is the case.
Performing an A/B testing type of analysis can only tell us WTA is the best in some limited benchmarks. However, what would really make this framework principle and shine are the insights and messages we can learn from your solution.

I further thought about the phenomenon and came up with some hypotheses. For example, WTA is an LSH that the collision probability of h(x) and h(y) corresponds to the ranking/ order alignment of $x_0...x_{d-1}$ and $y_0...y_{d-1}$. Because the choice of K is usually 2,4,8,16, it is more like local ranking information within a few random dimensions, while RS in Han et al is comparing the ranking of x and y globally (across d dimensions). Also, if you see your table 7,  when k goes larger, your performance saturates and degrades. If you start by analyzing the fundamental difference between RS and WTA along with your appendix B and C, you can maybe hypothesize that local ranking (among random dimensions) is more effective than global ranking and verify it with further experiments.  E.g. Intuitively with simple calculation, the half of the pair-wise order information among 4 elements (K=4) could be preserved with WTA hash code ( it is the index of the max so if the hash code is 0, it records 3 out of 6 relations: element 0 > elem1, >elem 2, >elem 3).

---

> ### Author Response · Authors · 2020-11-21
> **Answer to Reviewer 3**
>
> We would like to thank the reviewer for the comments and helpful suggestions.
>
> > Q: The presentation of contribution two is not convincing in two ways. First, at the algorithm level, if WTA is not the paper’s contribution, I recommend authors either weaken the way they describe it in abstract/intro and shorten sec 3.2; or explain in details why not other LSH variants or even learning to hash variants for unsupervised/semi-supervised setting? The reasoning in 3.2 is mostly adapted from the original WTA paper and also the follow-up best paper in CVPR[1]. Second, in experiments, I can see when changing from WTA to Simhash, the performance drops sharply (even worse than the baseline). Does it mean that WTA is the real component that makes the framework working? I can also infer it from table 1 that when BCE is omitted, the performance is affected most
>
> > A: Thanks for pointing this out.
> We followed the suggestion to update our description about WTA and revised section 3.2 to avoid confusion of considering proposing WTA as our novelty.
> The effectiveness of WTA is clearly demonstrated by [1]. The analysis of the benefits for WTA is inherent from the original paper. Moreover, we also included intuitive discussion on the potential benefit of taking into account the holistic structural information for WTA, rather than focusing on single local regions (like RS[2]). The role of WTA in our framework is to provide pair-wise pseudo labels on the unlabelled data for training. In Table 1, when BCE is omitted, thus the clustering branch of our framework is totally disablled, hence it is unable to perform clustering. In appendix C, we replace WTA by other alternatives. It can be seen, with other alternatives, our method can still discover novel categories. However, WTA appears to be more effective than others.
> The main novelty of our paper is the end-to-end framework for novel category discovery on multi-modal data, in which we extend the contrastive learning for both category- and instance-level discrimination on multi-modal data.
> WTA is employed at training time of our framework to provide pseudo labels for unlabelled data.
> This problem setting is still at its infancy. To our knowledge, this is the first framework that allows novel category discovery on multi-modal data.
> Further, we experimented on RS [2] by only replacing the ranking statistics with WTA, and didn’t observe performance gain. We added the results in appendix A. The results show that the other components of our approach are also important (see Table 1).
>
> > Q: The framework is ad hoc. There are many losses and design options making the practicality of the framework doubtful.
>
> > A: Our framework draws inspiration from prior success on other relevant problems. The model is end-to-end trainable. Different loss terms are for different purpose with clear motivations: (1) CE loss for supervised learning on labelled data; (2)BCE loss for clustering on unlabelled data; (3)CL loss for representation learning; (4) MSE loss is a common practice from the literature of semi-supervised learning (SSL). The ramp-up weighting is borrowed from SSL as well. Hence, we believe our model is generic and applicable for many real applications. Meanwhile, our code will be released upon publication. Any datasets of interest can be verified with our approach. We hope this work could be helpful to facilitate the future research for novel category discovery.
>
> > Q: In the experiments section, I recommend first presenting the final results first and then ablations.
>
> > A: Thanks. We agree that it is a good idea to put ablation study in the end, and would like to move the ablation to the last. However, moving it now will lead to table ID changes, which may confuse other reviewers for the tables they were referring to. We will update this in the final version of the paper as you suggested.
>
> ====
>
> [1] Dean et al. Fast, accurate detection of 100,000 object classes on a single machine.
>
> [2] Han et al. Automatically discovering and learning new visual categories with ranking statistics.

---

### Official Review · AnonReviewer5 · 2020-11-06
**Reasonable approach but needs clarification**

**Rating:** 5
**Confidence:** 3

**Review:**

SUMMARY

The paper proposes a multi-task loss for semi-supervised representation learning and category discovery (i.e. clustering unlabeled examples). At a high level, this paper adds a contrastive loss term to the approach from Han et al. 2020. The loss has five components: (i) an InfoNCE loss where positive pairs are transformations of the same image, (ii) an InfoNCE loss where positive pairs are images with the same class label, (iii) an MSE loss between the representations of transformations of the same image, (iv) a categorical cross-entropy loss for labeled data, and (v) a binary cross-entropy loss ("same class" vs. "different class") where class labels are based on thresholding a measure of similarity between representations (distance between Winner-Take-All hash codes). The loss supports multi-modal data in that (i) and (ii) may each be computed based on representations of any modality (intra-modal, e.g. anchor and positive are both audio) or any pair of modalities (inter-modal, e.g. anchor is audio and positive is video). The method is evaluated on category discovery benchmarks based on image and video datasets.


STRENGTHS

I think one can make a good argument that there has been too much focus on purely unsupervised methods lately - certainly the semi-supervised scenario this paper considers is far more realistic.

The paper is generally readable and the figures and tables are clear. The proposed method is reasonable and the empirical performance is strong. Most of the reported numbers are based on multiple runs, and an ablation study is included.

The analysis of multi-modal configurations in Table 2 is interesting and makes a clear case for cross-modal representation learning.

WEAKNESSES

It looks like the baseline methods were not re-implemented (the numbers were just copied over from previous work). Are all hyperparameters, models, etc. similar enough that these numbers can be fairly compared? Please clarify.

On the topic of fair comparisons, Han et al. 2020 seems to report stronger performance than is credited to them in Table 3. Also, the proposed method is very closely related to Han et al. 2020 (this paper adds a term to the loss proposed in that paper) but that is not clear from the text. Please make that more explicit.

There doesn't seem to be any description of validation sets or hyperparameter tuning procedures. Without this information, it's difficult to evaluate the significance of the quantitative results.

The paper would benefit from a bit more editing - see "minor comments" section below. In addition, the general flow of the writing could be improved. For instance, phrases "the accuracy is increased to X/Y/Z on dataset A/B/C" are used often and are a bit hard to parse.

OVERALL

The paper addresses an interesting and important question and presents strong results. However, there are some concerns regarding (i) how hyperparameters were tuned and (ii) the difference between the contributions of this paper and others, notably Han et al. 2020.

MINOR COMMENTS

**Abstract**

The terminology "We further introduce Winner-Take-All (WTA) hashing..." makes it sound a bit like WTA is something introduced in this paper.

**Section 3.1**

The notation $B \sim D^u \cup D^l$ is a bit unclear. Consider writing something like "$x_i \sim \mathrm{Unif}(D^u \cup D^l)$ for all elements $x_i$ in batch $B$."

Perhaps $Q(i)$ should be defined as $\{ q \in \mathcal{N} \backslash i : y_q = y_i\}$? Is $Q(i)$ meant to exclude only example $i$, or does it also exclude $z_{i'}$? Perhaps $p$ is meant to denote that?

The phrase "Without loss of generality" is a little out of place since no theoretical claim is being made.

**Section 3.2**

In the paragraph after Equation 4, "uisng" should be "using".

It's not really clear what motivates the use of WTA here - one could imagine all sorts of schemes for deciding whether two feature vectors are "similar enough" and this one seems a bit unusual. While there are additional comparisons in the appendix, it's hard to evaluate those results without knowing more about the hyperparameter selection procedure.

It's also not clear why the binary cross-entropy loss is chosen over e.g. yet another contrastive loss, where the pseudo-labels are used to determine whether two examples are to be considered similar or different.

Presumably $\hat{z_i}$ should be $\bar{z_i}$ after Equation 5.

Why limit this step to just unlabeled examples in each batch? It doesn't seem like there's any reason it couldn't also be applied to the labeled examples.

**Section 3.3**

Please explicitly define $\mathcal{L}^\mathrm{MSE}$. Does "the predictions" mean "the representations of $x_i$ and $t(x_i)$" in this context?

It seems like there's some missing text in the phrase "We set $(1-\omega(r))$ for contrastive learning."

**Section 4**

Is $e$ an arbitrary element of $\mathcal{P}(C^u)$ or is it the optimal permutation? The notation and text contradict each other.

Both $M$ and $U$ have been used to denote "the number of unlabeled instances" - it might be helpful to choose just one.

How is $C^u$ set? Is to chosen to be the correct number of classes? It seems like it would be appropriate to examine mis-specification of this parameter since it is generally not known in the "open world" setting the authors discuss. This is particularly important since $C^u$ is a hyperparameter of the proposed method (not just used for an evaluation metric).

**Table 1**

"Kinetcis-400" should be "Kinetics-400"

**Conclusion**

I think "imperially" should be "empirically".

**Update: After considering the other reviews and subsequent discussion, I have decided to maintain my score. As covered elsewhere in the discussion, the key shortcoming of the paper in its current form is that it provides little insight into why the proposed method outperforms the highly similar method in Han et al. For instance, the fact that WTA is crucial for the method proposed in the paper but does nothing for Han et al. is rather mysterious. I think it is necessary to provide more insight into these differences before readers can have full confidence in the proposed method.**

---

> ### Author Response · Authors · 2020-11-21
> **Answer to Reviewer 5  [Part 1]**
>
> We would like to thank the reviewer for the very detailed comments and helpful suggestions. We revised paper carefully and accordingly.
>
> > Q: the difference between the contributions of this paper and others, notably Han et al. 2020.
>
> > A: We revised papers following the suggestions to highlight our contributions. To sum up, our approach differs from RS [2] in many important aspects.
> (1) Our framework is end-to-end trainable, while RS requires three training stages. This design avoids our model to be biased towards labelled data throughout the whole training process while effectively transferring knowledge from the labelled data to the unlabelled data.
> (2) Our model is a generic framework that can deal with single- or multi-modal data. RS(Han et al. 2020) is not applicable to multi-modal data, as all the components are designed for images, while our framework can deal with multi-modal data. We applied the RS [2] on multi-modal video datasets, and its performance largely lags behind our framework (see Table 4).
> (3) We extend to contrastive learning to handle both category- and instance-level discrimination to learn robust representation for novel category discovery. RS [2] adopts the rotation prediction pretext-task, which is only applicable to natural images and is mainly able to capture lower level features.
> (4) We employ WTA in our framework for robust pairwise pseudo label generation for training on unlabelled data. The ranking statistics in RS [2] considers only the topk elements in the feature representation, while WTA considers multiple partial orderings, allowing the model to take the holistic structure of the object into consideration during comparison. Meanwhile topk struggles on multi-modal data. We hypothesis this is because the features of videos are much more complicated than the images, thus simply verifying the most salient regions in the feature vectors is not robust enough to compare new classes in the videos.
>
> >Q: It looks like the baseline methods were not re-implemented (the numbers were just copied over from previous work). Are all hyperparameters, models, etc. similar enough that these numbers can be fairly compared? Please clarify.
>
> > A: We tried to ensure fairness as much as possible. In particular, we used the official public code for the baseline experiments of KCL, MCL, DTC, and RS.  All the experiments used the same data splits and same model architectures, but still different methods have some different components. We used the same hyperparameters for the shared components. Among them, RS is the one that is most similar to ours.  Hence, to keep consistency, we reported the same values as in RS. We further extended RS framework to multi-modal tasks in table 4, and also enhanced it by replacing the original RotNet pretraining with our improved contrastive learning, using the same hyperparameters as our framework.
>
> > Q: On the topic of fair comparisons, Han et al. 2020 seems to report stronger performance than is credited to them in Table 3. Also, the proposed method is very closely related to Han et al. 2020 (this paper adds a term to the loss proposed in that paper) but that is not clear from the text. Please make that more explicit.
>
> > A: Thank you. For the RS [2] results, we carelessly copied the wrong line of results (the one without self-supervision) in Table 1 of RS [2] paper. We carefully updated the numbers for RS [2] in row (5) in table 3. We also reported the results of RS [2] in row (6) with an extra incremental learning scheme. After this update, the conclusion remains the same: our method significantly outperforms RS.
>
> > Q: There doesn't seem to be any description of validation sets or hyperparameter tuning procedures. Without this information, it's difficult to evaluate the significance of the quantitative results.
>
> > A: We added our WTA hyper parameter tuning procedures in appendix B.  We run experiments on labelled subset and keep the unlabelled subset untouched in these experiments. More specifically, we further split labelled subset into a smaller labelled set and a valid unlabelled set. While sweeping on different parameters is very expensive, so we run on one parameter and fix others every time.  We start from empirical value in [1], e.g, set WTA k=4, and sweep around its neighbourhoods. We find the results are generally stable when mu > 200 and 4<=k<= 8.
>
> > Q: The terminology "We further introduce Winner-Take-All (WTA) hashing..." makes it sound a bit like WTA is something introduced in this paper.
>
> > A: To avoid confusion, we have changed ‘introduce’ to 'employ’.
>
> > Q: The notation B∼Du∪D is a bit unclear. Consider writing something like "xi∼Unif(Du∪Dl) for all elements xi in batch B."
>
> > A: Thanks! We have updated it following the suggestion.

---

> > ### Author Response · Authors · 2020-11-21
> > **Answer to Reviewer 5 [Part 2]**
> >
> > > Q: Perhaps Q(i) should be defined as q∈N∖i:yq=yi? Is Q(i) meant to exclude only example i, or does it also exclude zi′? Perhaps p is meant to denote that?
> >
> > > A: Thanks for pointing out. Yes. It is meant to exclude only example i. We have corrected it.
> >
> > > Q: The phrase "Without loss of generality" is a little out of place since no theoretical claim is being made.
> >
> > > A: Thanks! We have changed it to ‘Therefore'
> >
> > > Q: In the paragraph after Equation 4, "uisng" should be "using".
> >
> > > A: Thanks! Fixed.
> >
> > > Q: It's not really clear what motivates the use of WTA here - one could imagine all sorts of schemes for deciding whether two feature vectors are "similar enough" and this one seems a bit unusual. While there are additional comparisons in the appendix, it's hard to evaluate those results without knowing more about the hyperparameter selection procedure.
> >
> > > A: The scheme for deciding the similarity is essential to the algorithm. One may consider cosine similarity, the ranking statistics in RS[2], or the NN as we experimented in the appendix. However, each of them has their pros and cons. Both cosine similarity and NN compare all dimensions in the feature vector, which is too strict compared with RS which only compares the most salient parts of the objects. Meanwhile, cosine and NN depend on the precise values in the high-dimensional feature vectors. As discussed in [1], the relative magnitude matters more when comparing high-dimensional feature vectors. This inspired us to use the ranking based method. However, though RS has been shown to be robust to noise for comparing images. We found it doesn’t work well for the multi-modal setting. We hypothesise this is because the features of videos are much more complicated than the images, thus simply verifying the most salient regions in the feature vectors is not robust enough to compare new classes in the videos. WTA has the merit of relative order comparison as well as capturing the structure information by comparing multiple partial orders, resulting in a more robust measure to compare complex information in the videos. Hence, we employ WTA in our framework. For the hyper parameter setting method, we have added the discussion in appendix B in the revised paper. We have further clarified this in the revised Section 3.2.
> >
> > > Q: It's also not clear why the binary cross-entropy loss is chosen over e.g. yet another contrastive loss, where the pseudo-labels are used to determine whether two examples are to be considered similar or different.
> >
> > > A: Contrastive learning focuses on representation learning, while the binary cross-entropy aims at learning to predict clustering assignment directly, given that we assume the number of classes in the unlabelled data is known. It has been proved in [3] that when the output dimension is the same as the class number, the optimally trained model will give a one-hot vector in the soft-max normalized output. Therefore, the cluster assignment for an unlabelled data point can be retrieved by argmax. In contrast, the output dimension of contrastive learning is not relevant to the number of classes, and the cluster assignment cannot be directly obtained after learning. Hence, using the binary cross-entropy can effectively produce the cluster assignment after training. We have further clarified this in the revised paper.
> >
> > > Q: Presumably zi^ should be zi¯after Equation 5.
> >
> > > A: Thank you. Fixed.
> >
> > > Q: Why limit this step to just unlabeled examples in each batch? It doesn't seem like there's any reason it couldn't also be applied to the labeled examples.
> >
> > > A: For labeled examples we can generate binary labels based on the ground-truth class labels. We've tried that but unfortunately did not see any improvement. The binary labels provide weaker supervision signals than multi-class labels, therefore, training with cross-entropy loss using multi-class labels is a better choice for the labelled data.
> >
> > > Q: Please explicitly define LMSE. Does "the predictions" mean "the representations of xi and t(xi)" in this context?
> >
> > > A: It refers to class prediction, i.e., softmax output, of a given input. We have explicitly defined this loss term in the revised paper.
> >
> > > Q: It seems like there's some missing text in the phrase "We set (1−ω(r)) for contrastive learning."
> >
> > > A: To make it more clear, we change to "we set (1−ω(r)) as the weight for contrastive learning"
> >
> > > Q: Is e an arbitrary element of P(Cu) or is it the optimal permutation? The notation and text contradict each other.
> >
> > > A: Thanks for pointing it out. e  is an arbitrary element of P(C^u). We denote the optimal permutation as e^*, which is obtained by Hungarian algorithm,  in the revised paper.

---

> > > ### Author Response · Authors · 2020-11-21
> > > **Answer to Reviewer 5 [Part 3]**
> > >
> > > > Q: Both M and U have been used to denote "the number of unlabeled instances" - it might be helpful to choose just one.
> > >
> > > > A: We used M to denote the number of unlabelled instances in a particular mini-batch, and U to denote the number of unlabelled instances in the whole dataset. We further clarified this in the revised paper.
> > >
> > > > Q: How is Cu set? Is to chosen to be the correct number of classes? It seems like it would be appropriate to examine mis-specification of this parameter since it is generally not known in the "open world" setting the authors discuss. This is particularly important since Cu is a hyperparameter of the proposed method (not just used for an evaluation metric).
> > >
> > > > A: Following RS[2], we assume the number of the classes, C^u,  in the unlabelled data is known a-priori. We have further emphasized this at the beginning of the method section. When C^u is not known, we can use the method introduced in DTC[4] to estimate C^u first, and then substitute the estimated number into our framework. We evaluated the performance of our approach on ImageNet using the unknown category numbers estimated by DTC. The estimates are 34/32/31 and the ground-truth numbers are 30/30/30 on the three unlabelled subsets. The average accuracy over three subsets is 84.1% which outperforms the previous SOTA RS[2] by 3.6%.  We have added this result in appendix D.
> > >
> > > > Q: * Table 1 *"Kinetcis-400" should be "Kinetics-400"
> > >
> > > > A: Thank you. Fixed.
> > >
> > > > Q: I think "imperially" should be "empirically".
> > >
> > > > A: Thank you. Fixed.
> > >
> > > ==========
> > >
> > > [1] Yagni et al. The power of comparative reasoning, ICCV 2011
> > >
> > > [2] Han et al. Automatically discovering and learning new visual categories with ranking statistics, ICLR 2020
> > >
> > > [3] Chang et al. Deep Adaptive Image Clustering, ICCV 2017
> > >
> > > [4] Han et al. Learning to discover novel visual categories via deep transfer clustering, ICCV 2019

---

### Decision · Program_Chairs · 2021-01-07
**Final Decision**

**Decision:**

Reject

**Comment:**

The reviewers unanimously raised concerns on the lack of insights on why the proposed method works better than Han et al., 2020, and why WTA brings significant gains only to the proposed method and not to Han et al. I think the paper is promising but providing these insights are critical to making the work convincing to the readers. The reviewers have made excellent points to improve the paper; I'd recommend the authors to incorporate them in their future submission.